# Causal Intervention-based Few-Shot Named Entity Recognition

**Zhen Yang,    Yongbin Liu,*    Chunping Ouyang**
University of South China
{uscyz094, yongbinliu03}@gmail.com, ouyangcp@126.com

## Abstract

Few-shot named entity recognition (NER) systems aim to recognize new classes of entities with limited labeled samples. However, these systems face a significant challenge of overfitting compared to tasks with abundant samples. This overfitting is mainly caused by the spurious correlation resulting from the bias in selecting a few samples. To address this issue, we propose a causal intervention-based few-shot NER method in this paper. Our method, based on the prototypical network, intervenes in the context to block the backdoor path between context and label. In the one-shot scenario, where no additional context is available for intervention, we employ incremental learning to intervene on the prototype, which also helps mitigate catastrophic forgetting. Our experiments on various benchmarks demonstrate that our approach achieves new state-of-the-art results.

## 1 Introduction

Named entity recognition (NER) is a fundamental task in information extraction, which involves identifying and classifying named entities in unstructured text. Several methods (Chiu and Nichols, 2016; Ma and Hovy, 2016; Lample et al., 2016; Peters et al., 2017) have been developed to achieve efficient results in NER.

In practical applications, few-shot named entity recognition has gained significant attention due to challenges in label collection and the high cost of manual labeling. Recent years have witnessed numerous studies focusing on few-shot NER. The current approaches primarily revolve around metric learning models, particularly prototypical networks. These models compute a prototype representation (Snell et al., 2017) for each class and assign labels to query samples based on their distance to the prototypes of each class (Fritzler et al., 2019; Yang

---
*Corresponding Author.

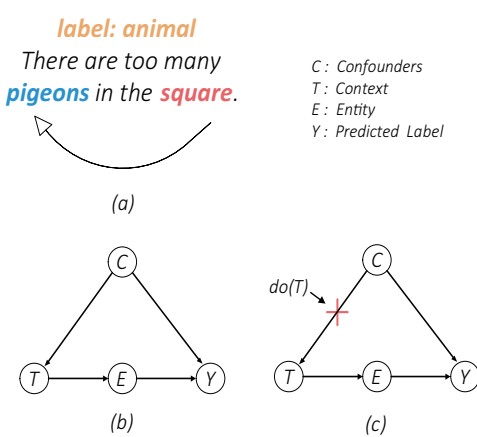

Figure 1: (a) An example of spurious correlation. *"Pigeons"* are easily associated with *"square"*, but not all animals are associated with *"square"*. (b) Causal graph of the example. Contexts $T$ ( such as the *"square"*), entities $E$ (such as *"pigeons"*), and class label $Y$ (such as the *"animal"* class type), $C$ means confounders brought by samples selection bias in the few-shot task. (c) Causal graph after *do*-operation

and Katiyar, 2020; Hou et al., 2020). Moreover, span-based methods (Wang et al., 2021a; Yu et al., 2021; Ma et al., 2022) have emerged to facilitate the accurate identification of entity boundaries in few-shot named entity recognition.

However, these existing methods tend to overlook the issue of overfitting caused by spurious correlation in few-shot tasks. While this problem is less prominent in tasks with abundant samples, it becomes crucial to address it in few-shot scenarios due to the bias introduced by the limited sample selection. As illustrated in Figure 1(a), we provide an example of spurious correlation. In this particular example, the context *"square"* is linked to the entity *"pigeon"* and the entity *"pigeon"* is associated with the label *"animal"*. It is worth noting that in some NER tasks, there may be cases where the *"animal"* label is mistakenly associated with the semantic feature of *"square"*, but this correlation does not apply to all animal entities.

The aforementioned issues can be attributed to the overfitting of the model to confounders present in the limited number of samples. Specifically, the few samples selection bias acts as a confounder that leads the few-shot NER model to learn spurious correlations between contexts and labels. For instance, in Figure 1(a), the context *"square"* becomes associated with the ground-truth entity *"pigeon"*. This denoted as $P(E|T)$, tends to create a positive association between the context and certain entities. When encountering the context *"square"*, it increases the likelihood that it belongs to the entity *"pigeon"*. However, this association based on the confounder misguides the model to associate the non-causal but positively correlated context $T$ with the class label $Y$, resulting in $P(Y|E,T)$. In other words, the context *"square"* is incorrectly perceived as a stable and intrinsic feature of the class label *"animal"*, even though it is a non-causal feature in the *"animal"* class.

Current metric-based methods fail to address the underlying confounders present in the few samples, particularly the few samples selection bias. In order to tackle the aforementioned challenges, we approach few-shot named entity recognition from a causal perspective. Figure 1(b) depicts the corresponding causal graph, which represents the causal relationships among the entity $E$, relevant context $T$, class label $Y$, and confounders $C$. The direct link between two nodes signifies the existence of causal relation between them.

In Figure 1(b), a backdoor path $T \leftarrow C \rightarrow Y$ is present, indicating that previous methods may erroneously learn spurious correlations caused by this backdoor path. To address this issue, we propose a model based on causal interventions. Specifically, we utilize context-based interventions to mitigate the spurious correlation between contexts and class labels. From a causal inference perspective, as depicted in Figure 1(c), we employ $P(E|do(T))$ instead of $P(E|T)$ to eliminate the influence of confounders. The *do*-operation, denoted as $do(T)$, allows us to establish the causal relationship between the contexts $T$ and the class labels $Y$ without the interference of confounders (Pearl, 2009). By calculating $P(Y|E,do(T))$, we can block the backdoor path and eliminate the spurious correlation arising from confounders. For the one-shot task, obtaining additional contexts is challenging. To overcome this, we draw inspiration from incremental learning (Thrun, 1995; French, 1999;

de Masson D'Autume et al., 2019) and employ the previous prototype to intervene in the current prototype to obtain the final prototype. This approach also prevents catastrophic forgetting (Thrun, 1995) in the model. Our contributions can be summarized as follows:

- From a causal perspective, we redefine the task and introduce context-based intervention for 5-shot. This intervention replaces the context and effectively prevents overfitting. By reducing the spurious correlation caused by selection bias in few shot samples, our method significantly enhances the model's generalization.

- For 1-shot, we use prototype-based intervention to reduce the spurious correlation between the current prototype and the label. We also apply sample reweighting to obtain more representative prototypes. Additionally, we utilize GPT for data augmentation to finetune the 1-shot data and further improve the accuracy.

- Our model was evaluated on the Few_NERD and SNIP datasets, and the results demonstrate its superiority, surpassing previous state-of-the-art methods. Through comprehensive experiments, we observed a significant improvement in generalization, with an average increase of 2% and up to 13% improvement across all tasks.

## 2 Related Work

### 2.1 Few-Shot Learning and Meta-Learning

Few-shot learning has gained significant popularity in natural language processing (NLP) research (Chen et al., 2019; Gao et al., 2020; Brown et al., 2020; Schick and Schütze, 2020; Lin et al., 2022). However, overfitting is a common challenge in few-shot learning. To mitigate this issue, researchers often incorporate source domain data (Han et al., 2018; Geng et al., 2019; Wang et al., 2021b). Initially, meta-learning techniques were widely applied in computer vision. The introduction of prototype networks led to the adoption of metric-based methods (Kulis et al., 2013; Vinyals et al., 2016; Snell et al., 2017). These methods involve encoding support set vectors, generating prototype representations, and calculating the distance between prototypes and query data using various metrics.

Finally, the query is classified based on the nearest prototype.

## 2.2 Few-shot Named Entity Recognition

Previous research in few-shot named entity recognition has explored various token-level approaches (Fritzler et al., 2019; Yang and Katiyar, 2020; Hou et al., 2020; Gong et al., 2021). One popular method is the use of prototype networks (Snell et al., 2017). Building upon this, NNShot and StructShot (Yang and Katiyar, 2020) introduced feature extraction and nearest neighbors techniques. The development of Few-NERD (Ding et al., 2021), a large-scale human-annotated few-shot NER dataset, allowed for the evaluation of ProtoBERT, NNShot, and StructShot methods. Additionally, prompt-based technologies (Cui et al., 2021) have emerged in this field. However, these methods have shown limited generalization capabilities. To improve the accuracy of few-shot named entity recognition, the method proposed in (Das et al., 2021) incorporates Gaussian embedding and contrast learning. Notably, these approaches often overlook the integrity and boundaries of entities. In response, span-level approaches (Wang et al., 2021a; Yu et al., 2021; Athiwaratkun et al., 2020; Wang et al., 2021c) have been proposed. For instance, ESD (Wang et al., 2021a) leverages span representation and matching to enhance entity recognition completeness. Similarly, Decomposed-MetaNER (Ma et al., 2022) fine-tunes parameters using support instances and focuses solely on entity localization during span detection.

However, the aforementioned methods primarily focus on the current support and query instances, neglecting the influence of context on entities. This oversight can result in the overfitting of both contexts and entities.

## 2.3 Causal Inference

Causal inference, as described by Pearl (2009), aims to eliminate confounders and obtain causal effects between variables, enabling accurate predictions in tasks. In light of the challenges posed by fitting issues and causal effects in few-sample scenarios, our objective is to leverage causal inference to enhance the robustness and transferability of our model. Many prior works have employed causal theory to improve model robustness, employing techniques such as counterfactual analysis, unbiased estimation, visual cues, and front-back door

adjustment to eliminate spurious correlations and identify causal effects between variables.

## 3 Methods

We give a few shot named entity recognition task descriptions in Appendix A. In the subsequent sections, we outline our method for few-shot named entity recognition.

### 3.1 Causal Intervention

The few-shot named entity recognition task is plagued by a significant selection bias in the data, leading to the emergence of spurious correlations that can mislead the model's overfitting. To address this issue, we propose a causal intervention-based approach. Figure 1 depicts the problem in the task, where $T$ represents the context, $E$ denotes the entity, $Y$ signifies the predicted label, and $C$ represents confounders, such as the few samples selection bias.

$T \rightarrow E \rightarrow Y$ The entity representation is learned in terms of contexts. The model uses the entity representation to get the predicted label.

$T \leftarrow C \rightarrow Y$ There exists a backdoor path in the model that creates a spurious correlation. The variable $C$ encompasses the few samples selection bias. These biases cause the context to overly focus on the current data, misleading the prediction of labels. This backdoor path results in a spurious correlation between the context and the predicted class label.

To address the backdoor path $T \leftarrow C \rightarrow Y$, we perform interventions on $T$. However, intervening on $T$ directly is challenging due to the confounders $C$ being difficult to capture. Therefore, we employ front-door adjustment in the calculation, as shown in Equation 1.

$$P(Y = y | do(T = t)) = \sum_E P(Y = y | E = e, do(T = t))$$
$$P(E = e | do(T = t)) \quad (1)$$

Based on the principles of causal inference, we have derived Eq 2 as the final result. The detailed derivation process can be found in Appendix B, providing a comprehensive explanation of the equation.

$$P(Y = y | do(T = t)) = \sum_E P(E = e | T = t)$$
$$\sum_{t'} P(Y = y | E = e, T = t') P(T = t') \quad (2)$$

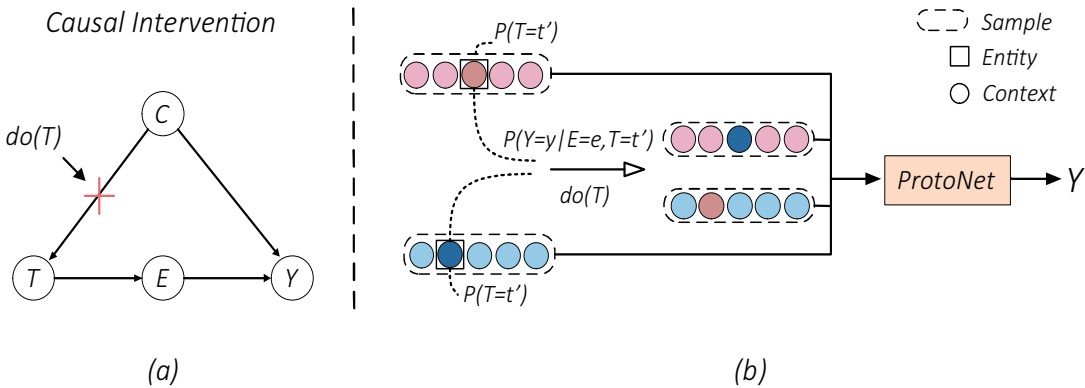

Figure 2: (a) The causal structure, where $T$ is the context, $E$ is the entity representation, $Y$ is the predicted label. Besides, there exists $C$, the confounders, such as samples selection bias. (b) Our context-based intervention, to block the spurious correlation between context and predicted label. We replace entities of the same type to get a new sample and feed it into the prototype network to get the predicted label.

Based on the three terms on the right of Eq 2, we introduce three important components: entity detection, context-based causal intervention, and sample reweighting. Each component has its specific role, and we will provide a detailed explanation of each below.

In the expression $\sum_E P(E = e | T = t)$, $e$ is the entity of the current sample. $t$ is the context of $e$. If we consider $E$ as a binary classification (entity vs. non-entity), this term denotes **entity detection**. It represents the probability of the entity $E$ being equal to $e$ given the context $T$ being equal to $t$.

In the expression $\sum_{t'} P(Y = y | E = e, T = t')$, $t'$ refers to the contexts of other entities that belong to the same type as e. This term indicates that to obtain the final prediction label $y$, we need to traverse $t'$ when $E$ is equal to $e$. So we propose the **context-based causal intervention** that involves replacing the contexts to observe the resulting predicted labels. This allows us to analyze the impact of different contexts on the prediction when the entity is fixed.

In the expression $P(T = t')$, with the calculation of the prototype, we propose the **sample reweighting** to take into account the weights assigned to different samples. This is done to address the potential differences between the source domain and the target domain distribution. By calculating the weights, the prototype is representative and takes into account the variations in the context distribution.

In the case of 1-shot, where no additional entities are available for intervention, we analyze the model. Figure 3 provides an illustration of the problem within the model. In this figure, $P$ denotes the

entity prototype, $Y$ represents the predicted label, and $C$ corresponds to the confounders.

$P \rightarrow Y$ The model calculates the Euclidean distance through the entity prototype and gets the predicted label.

$P \leftarrow C \rightarrow Y$ The presence of confounders, such as sample selection bias, can distort the calculation of the entity prototype, leading to a spurious correlation between the prototype and the predicted label. This spurious correlation, in turn, contributes to overfitting in the model.

To block the backdoor path $P \leftarrow C \rightarrow Y$, we use the intervention which is similar to $T$. The intervention on $P$ follows a similar approach in Eq 2.

In the 1-shot, where we don't have additional entities to intervene on, we focus on intervening on the prototype $P$. Since there is a direct causal path between $P$ and $Y$, we can simplify the intervention by directly modifying the prototype. Specifically, we propose the **prototype-based causal intervention** to consider both the previous prototype and the current prototype. Our experimental results confirms the importance of prototype intervention in improving the performance.

### 3.2 Entity Detection

In order to address the limitations of traditional prototype networks in capturing cross-class commonalities, we propose an entity detection mechanism. While the traditional prototype network averages the vectors of entities belonging to the same class to obtain class-specific prototypes, it overlooks the common features that may exist across different classes.

The entity detection approach, on the other hand,

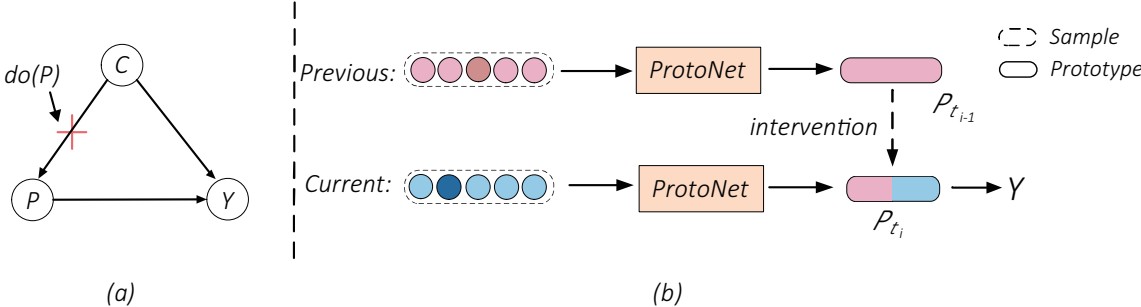

Figure 3: (a) Prototype-based intervention, where $P$ is the entity prototype, $Y$ is the predicted label, $C$ is the confounders. (b) We intervene in the prototype to block the spurious correlation between the prototype and the predicted label. During the current episode, the previous prototype is introduced to the current episode as knowledge. The two prototypes are combined as the current prototype. In the next episode, this prototype is introduced as the previous prototype.

aims to identify all named entities within the input sequence, without differentiating their specific classes. It takes into account the potential similarities between entities of all classes. To achieve this, we feed all the entities as a whole into the prototype network, as well as all the non-entities, to identify and locate all entities in the sequence. This approach allows us to uncover features that are common to all entities, which can be used for the initial filtering of the input.

When integrating the predicted labels, it is important to consider the dimensionality matching problem. Therefore, we expand the entity detection results to match the dimensionality of the class, ensuring compatibility for subsequent processing and classification tasks.

### 3.3 Context-based Causal Intervention

In the few-shot task, limited samples can lead to overfitting and hinder generalization. This is especially problematic when the entities are embedded within the contexts, as the model may overly adapt to specific context-entity relationships. To address this, we need to consider the causal relationships between entities, contexts, and predicted labels. Causal inference helps mitigate overfitting by removing spurious correlations and improving generalization.

In Figure 2(a), we present our approach for the few-shot named entity recognition task. The presence of confounders, such as the few samples selection bias, creates a backdoor path $T \leftarrow C \rightarrow Y$ between the context and predicted label. This backdoor path introduces spurious correlations, leading to the overfitting of both the context and the predicted label. To address this issue, our method

focuses on blocking the backdoor path. We achieve this by intervening on the context variable $T$, effectively blocking the influence of the confounders $C$ on the predicted label $Y$. We refer to this intervention strategy as context-based causal intervention.

The method outlined in Equation 2 involves iterating over each $T$ to calculate the final predicted label $Y$ for a given entity $E$. To accomplish this, we employ a traversal process illustrated in Figure 2(b). During training, for each sentence, we replace the entities in the sentence with other entities of the same type, one at a time, in order to explore all possible combinations. This traversal process allows us to consider the impact of different context-entity pairs on the predicted label. However, it's important to note that this process is only applied during training. During testing, we use the original data without any entity replacements.

### 3.4 Prototype-based Causal Intervention

In 1-shot experiments, context-based causal intervention is not possible due to the lack of additional samples. Instead, we use prototype-based intervention (Figure 3). Previous methods average support instances to calculate prototypes and rely on prototype-query distances for entity recognition. However, these methods can be influenced by confounders, leading to spurious correlations and overfitting.

To overcome this, we propose a prototype-based causal intervention. We intervene on prototypes ($P$) to block the $P \leftarrow C \rightarrow Y$ path, reducing spurious correlations. Our method incorporates prior knowledge by saving prior prototypes and combining them with the current class representation. This approach mitigates overfitting by considering both

current and previous data.

Prototype-based causal intervention effectively blocks spurious correlations between prototypes and predicted labels. It improves generalization and prevents overfitting in prototype calculation.

### 3.5 Sample Reweighting

We propose sample reweighting to address the issue of sample weighting. Instead of treating all samples equally, we assign different weights to each sample based on their relevance to the prototype calculation. This is done by calculating the distances between support samples and the query and using Softmax to determine the weights.

We also incorporate Maximum Mean Discrepancy (MMD) into the loss calculation to reduce the distribution difference between the training and test support data. The Softmax and MMD in Eq 3. By combining the MMD loss with the classification loss, we aim to improve domain adaptation.

Furthermore, we perform fine-tuning using the support instances from the test data to better adapt to the target domain. In the 1-shot scenario, where no additional data is available for fine-tuning, we use GPT for data enhancement before fine-tuning.

$$\alpha_i = softmax(h_\theta(x_q) - h_\theta(x_{s_i})) \qquad (3)$$

$$L(\theta) = \frac{1}{N} \sum_{i=1}^{N} CrossEntropy(y_i, h_\theta(\alpha_i x_i))$$
$$+ \sup_{\|f\|_H \leq 1} E_p[f(s)] - E_q[f(t)] \qquad (4)$$

Where $h$ is our network, $y$ is the true label, $p$ denotes the distribution of the source domain $s$, $q$ denotes the distribution of the target domain $t$.

## 4 Experiments

### 4.1 Dataset

Few-NERD (Ding et al., 2021) dataset[1] consists of two tasks: FewNERD-INTRA and FewNERD-INTER. FewNERD-INTRA has entities of different coarse-grained types in the training, validation, and test sets, while FewNERD-INTER allows overlapping coarse-grained types but disjoint fine-grained entity types. In FewNERD, the N-way K~2K shots approach is used, where N represents

the number of classes and K is the number of support examples per class. The dataset includes four settings: 5-way 1~2-shot, 5-way 5~10-shot, 10-way 1~2-shot, and 10-way 5~10-shot. We fine-tuned the model following (Ma et al., 2022) partition. However, we found that their dataset contains overlapping portions. To provide a better representation, we have included additional results after removing the overlaps, which are indicated in Table 1 with '†'.

SNIPS dataset (Coucke et al., 2018) focuses on slot filling and includes seven domains: D = {$D_1$, $D_2$, ..., $D_7$}. The leave-one-out strategy is followed, where a target domain is tested using a randomly selected validation domain, and the model is trained on the remaining source domains. In the few-shot slot-filling task of SNIPS, K examples per class are used in the support set. The task has 1-shot and 5-shot settings.

### 4.2 Parameter Settings

In our implementation, we utilized the Bert-base-uncased model (Devlin et al., 2018) as our base model. We set the maximum sequence length to 32. For optimization, we used AdamW (Loshchilov and Hutter, 2017). For further information on the parameter settings, please refer to Appendix C.

### 4.3 Evaluation metrics

For the evaluation of Few-NERD and SNIP, we followed ESD (Wang et al., 2021a), calculated the F1-score(F1).

### 4.4 Baselines

For a comprehensive comparison in Few_NERD, We select the top six state-of-the-art methods[2], such as ProtoBERT (Ding et al., 2021), NNShot (Ding et al., 2021), StructShot in Few-NERD (Ding et al., 2021), CONTAINER (Das et al., 2021), ESD (Wang et al., 2021a) and DecomposedMetaNER (Ma et al., 2022). To ensure a fair comparison, we reproduce DecomposedMetaNER using our maximum sequence length of 32. For SNIP, we choose the moethods including TransferBERT (Hou et al., 2020), MN+BERT (Hou et al., 2020), ProtoBERT (Hou et al., 2020), Ma2021 (Ma et al., 2021), L-TapNet+CDT (Hou et al., 2020), ESD (Wang et al., 2021a), Retriever (Yu et al., 2021) and ConVEx (Henderson and Vulić, 2020). The detailed baseline descriptions can be found in Appendix D.

---

[1]https://github.com/thunlp/Few-NERD

[2]https://paperswithcode.com/task/few-shot-ner

| Models | Intra | | | | | Inter | | | | |
|---|---|---|---|---|---|---|---|---|---|---|
| | 1~2-shot | | 5~10-shot | | Avg. | 1~2-shot | | 5~10-shot | | Avg. |
| | 5 way | 10 way | 5 way | 10 way | | 5 way | 10 way | 5 way | 10 way | |
| ProtoBERT* | $20.71_{\pm1.16}$ | $15.32_{\pm0.68}$ | $37.08_{\pm1.01}$ | $28.02_{\pm0.56}$ | 25.28 | $37.49_{\pm1.63}$ | $26.98_{\pm0.79}$ | $52.42_{\pm0.60}$ | $56.29_{\pm0.79}$ | 43.30 |
| NNShot* | $21.58_{\pm0.70}$ | $15.72_{\pm0.53}$ | $25.66_{\pm0.78}$ | $19.82_{\pm1.11}$ | 20.70 | $40.31_{\pm2.30}$ | $31.54_{\pm1.63}$ | $42.66_{\pm1.07}$ | $37.09_{\pm0.13}$ | 37.90 |
| StructShot* | $23.95_{\pm2.39}$ | $12.31_{\pm0.72}$ | $29.68_{\pm1.11}$ | $17.10_{\pm1.75}$ | 20.76 | $38.78_{\pm5.70}$ | $22.61_{\pm0.95}$ | $35.95_{\pm1.09}$ | $42.75_{\pm0.62}$ | 36.02 |
| CONTAINER | $40.00_{\pm0.71}$ | $35.89_{\pm0.16}$ | $54.28_{\pm0.64}$ | $48.48_{\pm0.69}$ | 44.66 | $52.17_{\pm2.74}$ | $50.00_{\pm1.46}$ | $61.11_{\pm0.76}$ | $59.84_{\pm0.36}$ | 55.78 |
| ESD | $37.12_{\pm0.89}$ | $31.26_{\pm0.53}$ | $50.50_{\pm1.79}$ | $33.38_{\pm4.71}$ | 38.07 | $57.56_{\pm2.52}$ | $52.89_{\pm1.11}$ | $69.92_{\pm0.56}$ | $66.90_{\pm0.60}$ | 61.82 |
| DecomposedMetaNER† | $45.06_{\pm1.38}$ | $39.55_{\pm0.73}$ | $55.08_{\pm1.71}$ | $48.26_{\pm2.25}$ | 46.99 | $61.74_{\pm0.13}$ | $57.40_{\pm1.77}$ | $66.20_{\pm0.58}$ | $61.38_{\pm0.15}$ | 61.68 |
| DecomposedMetaNER | $46.97_{\pm2.36}$ | $41.30_{\pm0.89}$ | $59.40_{\pm2.48}$ | $50.71_{\pm4.24}$ | 49.60 | $62.76_{\pm2.61}$ | $59.57_{\pm2.88}$ | $70.90_{\pm0.28}$ | $66.76_{\pm0.54}$ | 64.70 |
| **Ours†** | **$52.21_{\pm1.07}$** | **$41.47_{\pm0.98}$** | **$67.32_{\pm0.04}$** | **$59.78_{\pm0.23}$** | **55.20** | **$65.31_{\pm1.01}$** | **$59.91_{\pm0.11}$** | **$79.21_{\pm0.38}$** | **$73.47_{\pm0.59}$** | **69.48** |
| **Ours** | **$54.00_{\pm1.14}$** | **$43.58_{\pm1.20}$** | **$70.34_{\pm0.27}$** | **$61.33_{\pm0.47}$** | **57.31** | **$69.41_{\pm1.24}$** | **$60.67_{\pm0.73}$** | **$81.89_{\pm0.33}$** | **$75.86_{\pm0.10}$** | **71.96** |
| Ours+GPT-Aug | $73.72_{\pm2.64}$ | $62.61_{\pm1.33}$ | - | - | - | $80.85_{\pm2.14}$ | $67.63_{\pm0.59}$ | - | - | - |

Table 1: Performance of state-of-art models on Few-NERD. The "-" indicates that in order to have sufficient data fine-tuning, we use GPT for data augmentation (Ours+GPT-Aug) and generate an additional example. The "*" indicates that the results were directly obtained from the original paper. The '†' represents the experimental results obtained after removing the overlapping portions in the test set. The best results among the methods without using GPT are in **Bold**.

## 4.5 Results and Analysis

Our method achieves state-of-the-art results in both the FewNERD and SNIPS datasets, as shown in Table 1 and Table 2. In FewNERD, we observe significant improvements of 11% for the 5-shot setting and 1-8% for the 1-shot setting. These improvements demonstrate the effectiveness of our context-based causal intervention, entity detection, prototype-based causal intervention, and sample reweighting techniques. Similarly, in the SNIPS dataset, our method achieves an average improvement of approximately 13% for the 5-shot setting and an average improvement of 2% for the 1-shot setting. These results highlight the ability of our method to block spurious correlations, prevent overfitting, and improve overall performance. Meanwhile, GPT data enhancement improved our method by 2-20%. When we removed the overlapping portions, we observed that our results had a smaller decrease compared to DecomposedMetaNER, which further demonstrates the effectiveness of our model.

## 4.6 Ablation Study

The setting of the ablation study can be found in Appendix E. Table 3 demonstrates the contribution of each component in our method. Notably, we observed the following findings: Entity detection greatly enhances the effectiveness of context-based intervention, leading to an improvement of approximately 8% compared to not using entity detection in 5 shot. Context-based causal intervention helps resolve spurious correlation and leads to a 10% improvement. Prototype-based causal intervention is particularly effective in 1-shot scenarios, contributing to an 8% improvement. Sample reweighting enhances the model's performance by 4%. These results highlight the significance of each component in our proposed method.

## 4.7 Experimental Analysis

**How does entity detection improve entity recognition** Entity detection is essential for dealing with boundary cases. For example, it helps distinguish instances where *"the"* is part of an entity, like *"the Porcellian Club"* from cases where *"the"* is used as a common article, such as *"the year"*. Similarly, entity detection can correctly identify specific entities like *"American League"* while ignoring irrelevant words like airline names in phrases like *"American airline"*. By performing entity detection, the model can improve the accuracy.

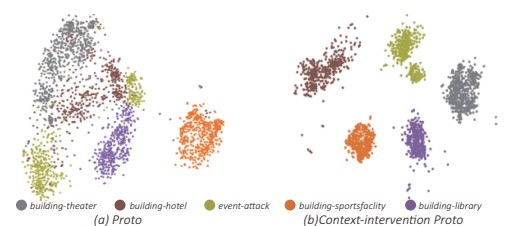

Figure 4: t-SNE visualization of entity replacing on Few-NERD Intra, 5-way 5~10-shot.

**How does context-based causal intervention improve entity recognition** In Figure 4, we provide t-SNE visualizations of the original prototype network and the prototype network with the context-based causal intervention. In the original

| | Models | We | Mu | Pl | Bo | Se | Re | Cr | Avg. |
|---|---|---|---|---|---|---|---|---|---|
| **1-shot** | TransferBERT | 55.82±2.75 | 38.01±1.74 | 45.65±2.02 | 31.63±5.32 | 21.96±3.98 | 41.79±3.81 | 38.53±7.42 | 39.06±3.86 |
| | MN+BERT | 21.74±4.60 | 10.68±1.07 | 39.71±1.81 | 58.15±0.68 | 24.21±1.20 | 32.88±0.64 | 69.66±1.68 | 36.72±1.67 |
| | ProtoBERT | 46.72±1.03 | 40.07±0.48 | 50.78±2.09 | 68.73±1.87 | 60.81±1.70 | 55.58±1.56 | 67.67±1.16 | 55.77±1.70 |
| | L-TapNet+CDT | 71.53±4.04 | 60.56±0.77 | 66.27±2.71 | 84.54±1.08 | 76.27±1.72 | 70.79±1.60 | 62.89±1.88 | 70.41±1.97 |
| | ESD | **78.25±1.50** | 54.74±1.02 | **71.15±1.55** | 71.45±1.38 | 67.85±0.75 | 71.52±0.98 | 78.14±1.46 | 70.44±0.47 |
| | **Ours** | 75.44±1.45 | **58.64±0.84** | 69.22±1.34 | **84.61±0.23** | **68.57±0.44** | **74.73±1.29** | **78.79±0.27** | **72.86±0.91** |
| | Ours+GPT-Aug | 73.30±1.17 | 53.56±0.17 | 88.98±0.44 | 84.62±0.53 | 70.22±0.39 | 73.52±0.61 | 76.57±0.28 | 74.40±0.13 |
| **5-shot** | TransferBERT | 59.41±0.30 | 42.00±2.83 | 46.07±4.32 | 20.74±4.36 | 28.20±0.29 | 67.75±1.28 | 58.61±3.67 | 46.11±2.29 |
| | MN+BERT | 36.67±3.64 | 33.67±6.12 | 52.60±2.84 | 69.09±2.36 | 38.42±4.06 | 33.28±2.99 | 72.10±1.48 | 47.98±3.36 |
| | ProtoBERT | 67.82±4.11 | 55.99±2.24 | 46.02±3.19 | 72.17±1.75 | 73.59±1.60 | 60.18±6.96 | 66.89±2.88 | 63.24±3.25 |
| | Retriever | 82.95(unk) | 61.74(unk) | 71.75(unk) | 81.65(unk) | 73.10(unk) | 79.54(unk) | 51.35(unk) | 71.72(unk) |
| | ConVEx | 71.50(unk) | 77.60(unk) | 79.00(unk) | 84.50(unk) | 84.00(unk) | 73.80(unk) | 67.40(unk) | 76.80(unk) |
| | Ma2021 | 89.39(unk) | 75.11(unk) | 77.18(unk) | 84.16(unk) | 73.53(unk) | 82.29(unk) | 72.51(unk) | 79.17(unk) |
| | L-TapNet+CDT | 71.64±3.62 | 67.16±2.97 | 75.88±1.51 | 84.38±2.81 | 82.58±2.12 | 70.05±1.61 | 73.41±2.61 | 75.01±2.46 |
| | ESD | 84.50±1.06 | 66.61±2.00 | 79.69±1.35 | 82.57±1.37 | 82.22±0.81 | 80.44±0.80 | 81.13±1.84 | 79.59±0.39 |
| | **Ours** | **94.67±1.33** | **90.34±0.45** | **91.73±0.82** | **94.70±1.55** | **94.75±1.82** | **93.18±1.63** | **89.75±0.28** | **92.73±0.11** |

Table 2: Below are the F1 scores with standard deviations on 7 domains of SNIPS. Please note that some methods do not provide deviation information in their paper (marked as 'unk'). The baselines for 1-shot and 5-shot settings differ because they did not report the 1-shot results in their paper.

| Setting | Entity Detection | Context-based Intervention | Sample Reweighting | Prototype-based Intervention | F1 score |
|---|---|---|---|---|---|
| | ✓ | ✗ | ✓ | ✓ | 54.00 |
| | ✓ | ✗ | ✗ | ✓ | 50.00 |
| 1∼2-shot | ✓ | ✗ | ✓ | ✗ | 46.09 |
| | ✗ | ✗ | ✓ | ✓ | 45.53 |
| | ✗ | ✗ | ✓ | ✗ | 43.82 |
| | ✗ | ✗ | ✗ | ✓ | 41.57 |
| 5∼10-shot | ✓ | ✓ | ✗ | ✗ | 70.34 |
| | ✗ | ✓ | ✗ | ✗ | 62.80 |
| | ✓ | ✗ | ✗ | ✗ | 60.44 |

Table 3: Ablation study: F1 scores on Few-NERD

prototype network, the embeddings are scattered and lack clear boundaries, making it prone to confusion. However, after applying the context-based causal intervention, the embeddings become more compact and exhibit clear boundaries. Entities of the same type are grouped closer together, while the distance between different types is increased. This optimization of the embedded space helps to distinguish and classify entities accurately.

**How does prototype-based causal intervention improve entity recognition** When analyzing the word *"German"* in the given query, the current prototype calculation may lead to misclassification. However, by intervening in the prototype and incorporating information from previous support data, we can improve the accuracy. For instance, if *"German"* was previously classified correctly as other-language, we can use that knowledge to re-

fine the current classification. By performing the prototype-based causal intervention, we can rectify the error and accurately classify *"German"* as other-language in the current query.

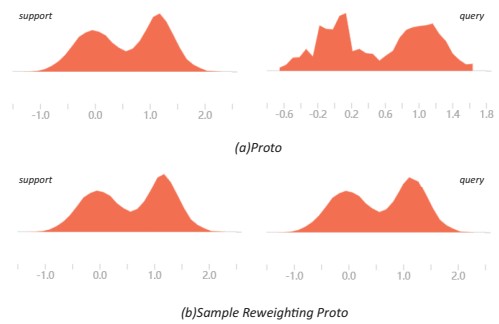

(a)Proto

(b)Sample Reweighting Proto

Figure 5: histogram of sample reweighting on Few-NERD Intra, 5-way 5∼10-shot.

**How does sample reweighting improve entity recognition** In Figure 5, we present histograms depicting the data distributions of the source and the target domain before and after applying sample reweighting. By comparing the histograms, we observe that after sample reweighting, the distributions of the source and target domains become more aligned, reducing the gap between them. This alignment is beneficial for the model's ability to transfer knowledge from the source domain to the target domain effectively.

## 5 Conclusion and Future Work

In this paper, we propose a method for few-shot named entity recognition that addresses the problem of overfitting by using causal interventions. We introduce entity detection, context-based and prototype-based interventions, and sample reweighting to improve performance. Our experiments show significant improvements in handling few-shot scenarios and transferring to new domains. Future work includes exploring dimensionality reduction techniques to reduce memory usage.

## Limitations

One limitation of our method is the increased memory usage due to context-based intervention. However, our method still offers an advantage compared to other methods that require a maximum length of 128, as we achieve good results with only 32. This helps mitigate the memory issue to some extent. In addition, compared to the significant improvement observed in the 5-shot, the boost for the 1-shot relatively smaller. We can use GPT. GPT data enhancement proves to be particularly helpful in improving performance in 1-shot scenarios. By leveraging GPT to enhance the limited amount of available data, we are able to achieve better results.

## Acknowledgements

The State Key Program of National Natural Science of China, Grant/Award Number:61533018; National Natural Science Foundation of China, Grant/Award Number: 61402220; The Philosophy and Social Science Foundation of Hunan Province, Grant/Award Number: 16YBA323; Natural Science Foundation of Hunan Province, Grant/Award Number: 2020JJ4525,2022JJ30495; Scientific Research Fund of Hunan Provincial Education Department, Grant/Award Number: 18B279,19A439,22A0316.

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

## A Task Formulation

Named Entity Recognition (NER) is commonly formulated as a sequence labeling problem, where given a sequence $\{x_1.. .x_n\}$, the goal is to assign a label to each $x_i$. These labels indicate whether $x_i$ belongs to a named entity category (e.g., person, location) or is not part of any entity (O class).

Few-shot NER aims to identify unknown entity classes using only a limited amount of training data, typically employing N-way K-shot learning. In this approach, episodes are iteratively constructed to define N-way K-shot training scenarios. During training, each episode consists of $N$ entity classes, with each class containing $K$ samples. This construction yields the support set, $S_{train} = \{(x^{(i)}, y^{(i)})\}_{i=1}^{N*Q}$. The query set is formed by sampling $Q$ samples from each of the $N$ classes, resulting in $Q_{train} = \{(x^{(i)}, y^{(i)})\}_{i=1}^{N*Q}$. It should be noted that $S_{train} \bigcap Q_{train} = \varnothing$.

The model is trained based on predictions made on the query set $Q_{train}$ during training. During testing, a few samples $S_{test}$ are used, and predictions are made for the corresponding query set $Q_{test}$. Similarly, $S_{test} \bigcap Q_{test} = \varnothing$. Importantly, the entity

classes present in the test set are not present in the training set, i.e., $Y_{train} \bigcap Y_{test} = \varnothing$.

We expect to block the backdoor path $T \leftarrow C \rightarrow Y$ and $P \leftarrow C \rightarrow Y$. So, we intervene on $T$ and $P$. First, when we intervene with T, considering $C$ as the confounders are hard to catch. Therefore, we use the front-door adjustment for the calculation, shown in the Eq 5. The intervention for P is similar.

## B    Causal Inference Formula

$$P(Y = y|do(T = t)) = \sum_E P(Y = y|do(T = t), E = e)$$
$$P(E = e|do(T = t)) \qquad (5)$$

Intervention of E, taking T and E as conditions, the conditional probability of Y is constant. So we get the Eq 6.

$$P(Y = y|do(T = t)) = \sum_E P(Y = y|do(T = t),$$
$$do(E = e))P(E = e|do(T = t)) \qquad (6)$$

Given that there is a direct causal path between T and E, the do-operation on T can be removed to get the Eq 7.

$$P(Y = y|do(T = t)) = \sum_E P(Y = y|do(T = t),$$
$$do(E = e))P(E = e|T = t) \qquad (7)$$

Meanwhile, considering that there is no direct causal path between T and Y. The do-operation on T will not affect the distribution of Y, so the formula can be further eliminated into Eq 8.

$$P(Y = y|do(T = t)) =$$
$$\sum_E P(Y = y|do(E = e))P(E = e|T = t) \quad (8)$$

Next, apply the total probability formula again to get the Eq 9.

$$P(Y = y|do(T = t)) = \sum_{t'} \sum_E P(Y = y|do(E = e),$$
$$T = t')P(T = t'|do(E = e))P(E = e|T = t) \qquad (9)$$

Reusing the previous direct causality between T and E, we get the Eq 10.

$$P(Y = y|do(T = t)) = \sum_{t'} \sum_E P(Y = y|do(E = e),$$
$$T = t')P(T = t')P(E = e|T = t) \qquad (10)$$

Similarly, according to the previous causality theorem, we can also remove the do operation on E, we get the final Eq 11.

$$P(Y = y|do(T = t)) = \sum_E P(E = e|T = t)$$
$$\sum_{t'} P(Y = y|E = e, T = t')P(T = t') \quad (11)$$

## C    Parameter Settings

Table 4: The hyperparameters of experiments

| Name | Value |
| --- | --- |
| Batch_size | 20 |
| Max_length | 32 |
| Learning rate | 1e-4 |
| Embedding dimension | 768 |
| Dropout | 0.1 |

## D    Baselline

**ProtoBERT** (Ding et al., 2021) uses BERT to get the vector representation of each token, and then averages all vectors of the same type as the the class representation according to the prototype network. Finally, calculate the distance between each category representation and query, and judge based on the nearest class.

**NNShot** (Ding et al., 2021) gets the feature representation for each token and calculates the distance between the query and each representation. Finally, the class is judged based on the nearest distance.

**StructShot** (Ding et al., 2021) adds an additional Viterbi decoder to the NNShot.

**CONTAINER** (Das et al., 2021) also uses BERT, and additionally uses contrast learning and Gaussian embedding to get the representation of each token. Then, fine-tuning on the support set and inference using the nearest neighbor method.

**ESD** (Wang et al., 2021a) uses inter and cross-span attention based on prototypes to get span representations. Also, it constructs multi-prototypes for O label.

**DecomposedMetaNER** (Ma et al., 2022) considers the few shot as a sequence labeling problem. MAML is used to initialize the model parameters, and meanwhile uses DecomposedMetaNER to find the optimal embedding space for entity recognition.

**TransferBERT** (Hou et al., 2020) is a model that applies BERT directly to few-shot sequence labeling through fine-tuning.

**Matching Net (MN)+BERT** (Hou et al., 2020) is a model that combines the Matching Network (MN) approach with BERT for token classification. It is similar to ProtoBERT but utilizes the matching network instead of prototype-based classification. This approach enhances the token classification task in few-shot sequence labeling.

**Ma2021** (Ma et al., 2021) approaches sequence labeling by formulating it as a machine reading comprehension problem. They propose a method that involves generating specific questions to extract slots from the query sentence. This approach allows for effective slot extraction in the sequence labeling task.

**L-TapNet+CDT** (Hou et al., 2020) is an advanced model that combines different techniques to improve classification in few-shot sequence labeling. It uses task-adaptive projection, pair-wise embedding, and collapsed dependency transfer mechanisms to enhance performance and capture task-specific information.

**Retriever** (Yu et al., 2021) is a retrieval-based method that performs classification by finding the most similar example in the support set. It uses a retrieval mechanism to identify the most relevant support instance and assigns the same label to the query based on this similarity.

**ConVEx** (Henderson and Vulić, 2020) is a model that follows a fine-tuning approach. It is initially pre-trained on the Reddit corpus using sequence labeling objective tasks and then fine-tuned on both the source and target domain annotated data to perform few-shot sequence labeling.

## E   Ablation study setting

In order to evaluate the contribution of the different components of the proposed method, we performed the following baseline as an ablation study: for 1∼2-shot, 1) We use the basic prototype network and do the entity detection, prototype-based intervention and sample reweighting. 2) Without sample reweighting, we use the basic prototype network and do the entity detection and prototypr-based intervention. 3) we use the entity detection and sample reweighting. 4) Without entity detection, we only use the basic prototype network and do sample reweighting and prototype-based intervention. 5) We only do the sample reweighting.

6) We only use prototype-based intervention. For 5∼10-shot, 1) Without sample reweighting, we make a context-based intervention and do the entity detection. 2) We only make a context-based intervention for the basic prototype network. 3) We only use the basic prototype network and do the entity detection.