# OpenReview forum: "Causal Intervention-based Few-Shot Named Entity Recognition"
_EMNLP/2023/Conference — EMNLP 2023 Findings_

### Official Review · Reviewer_c6Vr · 2023-07-24

**Typos Grammar Style And Presentation Improvements:** 1. In line 195, Reference “Pearl (200…
**Soundness:** 2

**Excitement:**

3: Ambivalent: It has merits (e.g., it reports state-of-the-art results, the idea is nice), but there are key weaknesses (e.g., it describes incremental work), and it can significantly benefit from another round of revision. However, I won't object to accepting it if my co-reviewers champion it.

**Missing References:**

There are some causal intervention approaches that need to be referenced.[1-3]
[1] Wu Y X, Wang X, Zhang A, et al. Discovering invariant rationales for graph neural networks[J]. arXiv preprint arXiv:2201.12872, 2022.
[2] Wu Q, Zhang H, Yan J, et al. Handling distribution shifts on graphs: An invariance perspective[J]. arXiv preprint arXiv:2202.02466, 2022.
[3] Fan S, Wang X, Mo Y, et al. Debiasing graph neural networks via learning disentangled causal substructure[J]. Advances in Neural Information Processing Systems, 2022, 35: 24934-24946.

**Paper Topic And Main Contributions:**

This paper uses context-based causal intervention and prototype-based causal intervention to address the overfitting problem in the few-shot NER task. For 1-shot, the author uses prototype-based intervention to reduce the spurious correlation between the current prototype and the label. For 5-shot, the author introduces context-based intervention to replace the context to prevent overfitting. The experiments on various benchmarks demonstrate that this approach achieves new state-of-the-art results.

**Questions For The Authors:**

A. In Section 1, the author says “However, these existing methods tend to overlook the issue of overfitting caused by spurious correlation in few-shot tasks.” What elements are involved in this spurious association?

B. In Section 4, The author only used GPT for data augmentation on the 1-shot setting, so what is the effect of using GPT for data augmentation on the 5-shot setting?

C. In Section 4.2, the author describes utilizing the Bert-base-uncased model as the base model. If the author replaces Bert with Roberta and others pre-trained language models, will better results be achieved?

D. In Section 4.7, the author just describes the word “German”, it is best to use visualization to demonstrate this case. And need give more description to analyze the reason.

E. Besides the pre-trained method, will it effectively improve the accuracy of other models if the papers’ method is applied to other few-shot NER methods?


**Reasons To Accept:**

1. This paper uses causal intervention to solve the overfitting problem caused by data selection bias in the few-shot scenario, the research motivation is reasonable.

2. The author designs two causal intervention approach, named context-based causal intervention for 5-shot scenario and prototype-based causal intervention for 1-shot scenario.

3. The author compared many novel few-shot NER methods in the experimental section and achieved SOTA results on the Few_NERD and SNIPS datasets, which have certain application values.

**Reasons To Reject:**

Insufficient in the experimental part:
1. Lack of case study analysis for more intuitive display.
2. The author only used GPT for data augmentation on the 1-shot setting, so what is the effect of using GPT for data augmentation on the 5-shot setting?
3. In Section 4.6, the author did not compare the effectiveness of using the sample reweighting method under the 5-shot setting.
4. In Section 4.7, the author just describes the word “German”, it is best to use visualization to demonstrate this case.

The paper needs further polishing and revision:
There are many formatting errors and inconsistent correspondence before and after the paper.

**Reproducibility:**

4: Could mostly reproduce the results, but there may be some variation because of sample variance or minor variations in their interpretation of the protocol or method.

**Reviewer Confidence:**

5: Positive that my evaluation is correct. I read the paper very carefully and I am very familiar with related work.

---

> ### Author Rebuttal · Authors · 2023-08-29
>
> 1.The response to 'Lack of case study analysis for more intuitive display‘.
>
>
>
> ​	In the results analysis section of the paper titled "How does prototype-based causal intervention improve entity recognition," we provide examples to illustrate the impact of our proposed method. **For further clarity, we have included an explanatory illustration below.**
>
>
>
> ​	Due to the webpage's configuration, it appears that we are unable to display the properly formatted example diagram at the moment. We will incorporate this section in the upcoming versions. Here, let me briefly describe the illustration. Regarding the term "German," if it is included in the current support data and categorized as the "O" class, then in the context of the current query data, a conventional prototype network would classify it as the "O" class as well. This illustrates that the model is only fitting to the immediate data and is forgetting previous data. However, when we incorporate prior knowledge, if the term "German" appeared in support data in previous rounds and was labeled as the "other-language" class, the model can correctly classify it as the "other-language" category. This is due to the fact that the previous prototypes retain relevant feature knowledge, which we reintroduce into the current scenario.
>
>
>
> ​	In addition to the above, to facilitate better understanding, **we have also provided additional examples to illustrate the functioning of the model. You can find these extra examples in our responses to the "Questions For The Authors-D" section that follows.**
>
>
>
> 2.The response to 'The author only used GPT for data augmentation on the 1-shot setting, so what is the effect of using GPT for data augmentation on the 5-shot setting'.
>
>
>
> ​	(1) To address your first question, the decision not to employ GPT-based data augmentation for the 5-shot setting stems from the fact that in 5-shot scenarios, there are already five sentences for each category. This constitutes a substantial amount of data. When applying context-based intervention in the 5-shot setup, the number of sentences increases significantly. Each sentence for a category can be swapped with the other four sentences within the same category, yielding a total of 25 sentences. **This dataset volume is already substantial for the model to learn from, eliminating the necessity for additional data augmentation.** **Moreover, our model has achieved excellent results in this context, affirming our decision not to employ GPT augmentation for the 5-shot scenario.**
>
>
>
> ​	(2) **However, your concerns are valid and insightful. Therefore, we have taken the time to conduct supplementary experiments.** Due to time constraints, we've performed these experiments solely on the FewNERD-inter dataset for the 5-way 5-shot scenario.
>
> （Supplement Table D1: Results in FewNERD-intra dataset 5-way 5-shot)
>
> | F1（%）  | FewNERD-intra |
> | -------- | ------------- |
> | SOTA     | 59.4          |
> | Ours     | 70.34         |
> | Ours+GPT | 71.08         |
>
> ​	**This indeed confirms the effectiveness of utilizing GPT for data augmentation. This clarification underscores our commitment to comprehensive experimentation and validation of our proposed approach.**
>
>
>
> 3.The response to 'In Section 4.6, the author did not compare the effectiveness of using the sample reweighting method under the 5-shot setting'.
>
> ​	**We did not employ sample reweighting in the 5-shot scenario. The reasons for not using it in the 5-shot context can be summarized as follows**:
>
>
>
> ​	(1) In the 5-shot setting, we utilized context-based intervention, where each sample's context was replaced with the context of other samples within the same category. This expansion resulted in an increased number of sentences per category, growing from 5 to 25 sentences. **This augmented dataset requires more memory to handle**. (It's worth noting that while the number of samples increased significantly, our sentence maximum length was set at 32, not the 128 used in SOTA methods, which mitigates memory challenges.) However, if we were to apply sample reweighting after expanding to 25 sentences, it would demand an even larger memory footprint. Therefore, we chose not to employ sample reweighting in the 5-shot setting and instead focused on scenarios with fewer samples, such as 1-shot tasks.
>
>
>
> ​	(2) We did experiment with combining context-based intervention and sample reweighting in the 5-shot setting on cards with larger memory capacity. **This did yield a modest improvement of 1-2 points in performance. However, the memory consumption was substantial, requiring a transition from 32GB to 80GB cards.** Although the 80GB cards were not fully utilized, the memory overhead remained significant. Given the marginal gain of 1-2 points in comparison to the extensive memory expenditure, we chose to forgo sample reweighting in the 5-shot scenario.
>
>
>
> 4.The response to 'In Section 4.7, the author just describes the word “German”, it is best to use visualization to demonstrate this case'.
>
>
>
> ​	**Due to the webpage constraints, it seems that we are unable to display the prepared example images. We will incorporate this section in the subsequent versions.** **Here, I'll briefly describe the images.** For the term "German," in the current set of support data, it is categorized as class "O." Consequently, in the current query data, if the term "German" appears, a conventional prototype network would classify it as class "O" as well. This illustrates that the model is only fitting to the current data and forgetting the previous data. However, when we integrate prior knowledge, in previous rounds, the support data had instances of the term "German" labeled as the "other-language" class. This enables the model to accurately classify it as the "other-language" category when it appears in the query data. This effect is due to the prior prototypes retaining pertinent feature knowledge, which is then incorporated into the present context.
>
>
>
> 5.The response to 'The paper needs further polishing and revision: There are many formatting errors and inconsistent correspondence before and after the paper'.
>
>
>
> ​	I appreciate your feedback, and we will certainly undertake thorough revisions and optimizations of both the grammar and content of the paper in subsequent versions. For the finalized published paper, we plan to enlist the assistance of native professional experts to further refine and polish the writing.
>
>
>
> **QA.** The response to 'A. In Section 1, the author says “However, these existing methods tend to overlook the issue of overfitting caused by spurious correlation in few-shot tasks.” What elements are involved in this spurious association'.
>
>
>
> ​	I appreciate your inquiry. In this context, we present an example as depicted in the figure below, which is also Figure 1 in the paper. In this example, "pigeons" belongs to the "animal" category, which is correct. However, in the same sentence, "pigeons" is situated closely to "square." This proximity may lead the model to erroneously associate "square" with the "animal" category during feature representation and classification. **While this phenomenon has a limited impact in larger datasets due to the volume of samples, it becomes more pronounced in few-shot scenarios. With fewer samples, the model overfits to this limited data, making such errors more conspicuous.**This association is misleading – not all entities mentioned alongside "square" should be categorized as "animal." **Therefore, we contend that a spurious correlation exists between context and labels, manifesting this false association.**
>
>
>
> **QB**.The response to 'B. In Section 4, The author only used GPT for data augmentation on the 1-shot setting, so what is the effect of using GPT for data augmentation on the 5-shot setting'.
>
>
>
> ​	Thank you for sharing your concerns, and we truly understand your queries. Let me explain why we didn't utilize GPT for 5-shot tasks and then provide the additional results we've obtained by using GPT for 5-shot settings.
>
>
>
> ​	(1) Initially, the rationale behind not employing GPT data augmentation for 5-shot tasks is due to the fact that each category already has 5 sentences in the 5-shot scenario. When context-based intervention is applied in 5-shot tasks, the number of sentences increases significantly. Each sentence for a category can be swapped with the other four sentences within the same category, resulting in a total of 25 sentences. **This dataset volume is already substantial for the model to learn from, making the introduction of additional augmented data unnecessary. Furthermore, we've achieved excellent results in this context, which is why we didn't adopt GPT augmentation for 5-shot tasks.** To further clarify, although our method increases the number of sentences and might appear to consume more memory, our sentence maximum length is only 32, yet we've achieved remarkable outcomes. Unlike current SOTA methods that require a maximum length of 128, we've significantly conserved memory in this regard.
>
>
>
> ​	(2) However, your concerns are indeed insightful. Therefore, we've taken the opportunity to conduct supplementary experiments. Due to time constraints, we've exclusively conducted these experiments on the FewNERD-intra dataset for the 5-way 5-shot scenario. You can see the Supplement Table D1 in response 2. This comprehensive clarification underscores our commitment to rigorously exploring different aspects of our approach and validating its effectiveness.
>
>
>
> **QC.** The response to 'C. In Section 4.2, the author describes utilizing the Bert-base-uncased model as the base model. If the author replaces Bert with Roberta and others pre-trained language models, will better results be achieved'.
>
>
>
> ​	Thank you for your insights, and your questions have indeed provided us with valuable perspectives. In the realm of vector representations, it's worth noting that BERT serves as a classic model but is interchangeable with other models. Due to time constraints, **our experimentation focused on Roberta**.
>
> （Supplement Table D2: Results in FewNERD-inter dataset 10-way 1-shot)
>
> | F1（%） | FewNERD-inter |
> | ------- | ------------- |
> | SOTA    | 60.47         |
> | Ours    | 60.27         |
>
> ​	 We preliminarily attribute this observation to the model's focus on data fitting. During the training process, both BERT and Roberta's parameters are fine-tuned, and across multiple training rounds, these parameters tend to converge towards the dataset, yielding relatively similar final vector representations.
>
>
>
> **QD**.The response to 'D. In Section 4.7, the author just describes the word “German”, it is best to use visualization to demonstrate this case. And need give more description to analyze the reason‘.
>
>
>
> ​	I appreciate your inquiries. Allow me to elucidate the purpose of introducing prototypes and subsequently provide more examples for illustration.
>
>
>
> ​	(1) **For the example of "German."** In few-shot named entity recognition tasks, the model learns from the current support data and predicts the category of entities in the query data. Given the minimal sample size per round in few-shot scenarios, the model tends to overfit the current data, leading to certain erroneous behaviors. When a word that exists in both support and query data is categorized as "O" (non-entity) in the support set, the model might wrongly classify the same word in the query as "O." This constitutes an inaccurate prediction for a word that should belong to an entity category. Fortunately, if that word was correctly categorized as an entity in previous rounds, the prototype representation will retain this information. By storing prototype knowledge from previous rounds and incorporating this knowledge in subsequent rounds, the model can correctly classify the word, thereby mitigating the model's forgetting of past knowledge and overfitting to the current data.
>
> ​	(2)  **We have introduced new examples to further illustrate this phenomenon.** For instance, consider the word "chocolate" in the diagram below. (Due to constraints in webpage settings, we are currently unable to display the diagram here. We will include this section in future versions. For now, let me provide a brief description of the diagram.) In the current support data, "chocolate" is classified as category O. Consequently, when it appears in the query data, it is also categorized as O. However, in reality, it should belong to the "product-food" category. By incorporating previous prototype knowledge, which includes instances where "chocolate" was categorized as "product-food" in earlier rounds, the model is able to correctly classify it as "product-food," thus rectifying the earlier misclassification.
>
>
>
> ​	These explanations and examples collectively underscore how the introduction of prototypes facilitates correct classification and addresses challenges in few-shot scenarios.
>
>
>
> **QE.** The response to 'E. Besides the pre-trained method, will it effectively improve the accuracy of other models if the papers’ method is applied to other few-shot NER methods'.
>
>
>
> ​	Thank you for your inquiry. **I'd like to start by clarifying that our proposed method is an independent approach, not a plug-in type of solution.** I will then explain that our method is based on improving prototype networks and has yielded significant enhancements. Below, I'll provide detailed explanations to address your concerns.
>
>
>
> ​	(1) **Our method consists of three main components**: entity detection, context-based causal intervention/prototype-based causal intervention, and sample reweighting. In the entity detection component, when dealing with five classes, the model doesn't treat the data as five specific classes plus an "other" class. Instead, it directly groups the five classes into an "entity" class, creating an "entity class" plus an "other class" setup. This step helps the model better understand the features of entity classes, aiding in the initial classification of entity categories in the sequence. In the context-based causal intervention part, for each entity, the model replaces the context of entities of the same type to reduce the influence of sample selection bias on the labels. In the case of 1-shot data, where each class has only one sentence, this type of context replacement is not feasible. Hence, we consider an alternative causal intervention approach, prototype-based causal intervention. The model can overfit the current data, which leads to excessive attention on the current prototypes. If there is bias in the current prototypes, it can affect the model's predictions on query data. To mitigate this, we introduce previous prototypes and combine them with the current ones. In practice, we store prototypes obtained by the previous prototype network, and when encountering prototypes of the same type later, we average them to form the feature representation for that type. Finally, in the sample reweighting part, we believe that different support samples should contribute differently to the prototypes. Support data more similar to the query should carry more weight. We calculate the similarity between the vector representations of support and query data, and assign different weights to calculate the prototypes. Overall, our proposed method is an independent approach rooted in prototype networks and causal inference ideas. It's not an application of a plug-in to other methods.
>
>
>
> ​	(2) It's worth noting that our method is based on prototype networks. By incorporating causal inference concepts into prototype networks, we construct both context-based and prototype-based causal interventions. In the ablation experiments, we compare results obtained using standard prototype networks and causal prototype networks. Table 3 indicates that causal inference significantly improves prototype networks. Figure 4 visually demonstrates that context-based causal intervention yields more compact and distinct entity categories compared to standard prototype networks. **These findings underscore how causal inference enhances prototype networks. Thus, we believe that applying causal inference concepts to other methods could also yield improvements.**
>
>
>
> ​	In conclusion, I hope my explanation has addressed your concerns.
>
>
>
> **Missing References.**
>
>
>
> ​	Thank you for your addition. In our upcoming versions, we will reference these sources and provide explanations accordingly.
>
>
>
> **Typos Grammar Style And Presentation Improvements**
>
>
>
> ​	First of all, thank you for your corrections. We will address these issues in subsequent versions to ensure greater compliance with standards. Regarding the example issue you raised, in various tasks, similar to the context you mentioned, background information can indeed lead to false associations with task labels. For instance, in common image-related problems, scenarios like pigeons being often present in squares are prevalent. **However, such associations are misleading since the presence of dogs on grass in images or horses being ridden by people in textual contexts doesn't inherently relate to the respective labels.** Below, we provide several papers that have been accepted at some top conferences to illustrate that these false associations exist both in natural language processing and computer vision domains.
>
>
>
> - Yang X, Zhang H, Qi G, et al. Causal attention for vision-language tasks[C]//Proceedings of the IEEE/CVF conference on computer vision and pattern recognition. 2021: 9847-9857.
> - Niu Y, Tang K, Zhang H, et al. Counterfactual vqa: A cause-effect look at language bias[C]//Proceedings of the IEEE/CVF Conference on Computer Vision and Pattern Recognition. 2021: 12700-12710.
> - Chang C H, Adam G A, Goldenberg A. Towards robust classification model by counterfactual and invariant data generation[C]//Proceedings of the IEEE/CVF Conference on Computer Vision and Pattern Recognition. 2021: 15212-15221.

---

### Official Review · Reviewer_56nu · 2023-07-31

**Soundness:** 3

**Excitement:**

4: Strong: This paper deepens the understanding of some phenomenon or lowers the barriers to an existing research direction.

**Paper Topic And Main Contributions:**

This paper proposes a mechanism for prototypical networks to help improve the correct labeling of entities in few-shot scenarios. It is claimed that this mechanism, a causal intervention, is based on causal analysis. The main aim of this mechanism is to remove a spurious dependency between context and predicted entity label (it is claimed that this spurious entity is due to a sample selection bias). The main contribution of this paper is to improve F1 scores on two datasets, namely Few-NERD (INTRA (1~2-shot, 5~10-shot), INTER(1~2-shot, 5~10-shot) and SNIPS (1-shot, 5-shot). Most of the results obtained surpass results from previous works.

**Questions For The Authors:**

1. In the tables of results, do +- indicate standard deviations across different random seeds?
2. There seem to be 2 mistakes in Table 2: on the Mu and Se domains, the model with highest score is L-TapNet+CDT and not the model proposed in this paper, is this correct?
3. Could you please address the issues from Reasons to Reject?


**Reasons To Accept:**

1.	The manuscript is clearly written
2.	Most of the results obtained surpass those from the literature, and in some cases by a wide margin.
3.	The solution seems to be computationally feasible and derivation of the mechanism proposed is simple
4.	Ablation experiments are provided to see the effect of each component in the proposed mechanism


**Reasons To Reject:**

There are several aspects that remain unclear:
1. The authors propose a causal model, in Fig 1, that aims to show the dependencies between predicted label, target entity, context, and the confounder (spurious factor due to sample selection bias). However, the way how a model captures the dependencies between such variables may not be causal. As a matter of fact, current models are statistical machines that capture statistical correlations. It is not clear if the models used here (prototype networks) are causal or not as it is not mentioned. Furthermore, even if the model could capture causal dependencies, that doesn't mean it is actually modeling such dependencies as proposed in Fig 1; i.e., there is no rationale, explanation, or supportive evidence for the validity of such a causal model.
2. The section on causal inference is too short and there is no deep elaboration on the connection between this theory and the work proposed in this paper. For example, under which grounds the causal model from Fig 1 is faithful to causal inference? (On the other hand, the authors could clearly eliminate many paragraphs from the paper that are highly repetitive.)
3. Derived from the point above, there is no experimental proof that supports the causal model proposed in this paper. Is the sample selection bias tested? Is there really a causal connection for the form T <- C -> Y?  Could the confounder be due to some other biases?
4. The mechanism proposed is derived from Eq. 1, which is unclear if this equation is faithful to the theory of causal inference (no citation nor explanation is given).
5. The mechanism proposed consists of some factors, some of which are slightly unclear. For example, for the factor called prototype-based intervention, the intervention done to block the effect of the confounder on the prediction needs the signal from the previous prototype vector. It is unclear what is the previous prototype, what is the exact operation of this intervention, and what is the rationale behind such an intervention. Similarly, for the factor called sample reweighting, it is unclear the logic behind reducing the distribution difference between training and test support data (based on common sense, this operation seems, actually, undesirable since test data should not be used at training time, only at test time).

**Reproducibility:**

3: Could reproduce the results with some difficulty. The settings of parameters are underspecified or subjectively determined; the training/evaluation data are not widely available.

**Reviewer Confidence:**

3: Pretty sure, but there's a chance I missed something. Although I have a good feel for this area in general, I did not carefully check the paper's details, e.g., the math, experimental design, or novelty.

**Typos Grammar Style And Presentation Improvements:**

It would be nice if the paper were less repetitive about the main point of the sample selection bias and showed more examples of the problem and how the mechanism proposed work on instances from the datasets used.

---

> ### Author Rebuttal · Authors · 2023-08-29
>
> 1.The response to 'The authors propose a causal model, in Fig 1, that aims to show the dependencies between predicted label, target entity, context, and the confounder (spurious factor due to sample selection bias). However, the way how a model captures the dependencies between such variables may not be causal. As a matter of fact, current models are statistical machines that capture statistical correlations. It is not clear if the models used here (prototype networks) are causal or not as it is not mentioned. Furthermore, even if the model could capture causal dependencies, that doesn't mean it is actually modeling such dependencies as proposed in Fig 1; i.e., there is no rationale, explanation, or supportive evidence for the validity of such a causal model.'
>
> ​	(1) First, I will explain the entire process of causal reasoning. In causal inference, **there is a very classic work**: Judea Pearl's "The Book of Why," which includes a detailed diagram of the causal inference process. **In summary, researchers propose hypotheses based on background knowledge and construct causal models.** For issues that can be answered by the model, i.e., those that can be mathematically formulated, statistical estimation methods (commonly employed in machine learning) can be used. If the model cannot answer the question, i.e., if it cannot be formalized, researchers return to the hypothesis stage and reconstruct the causal model. In this process, hypotheses correspond to mathematical deduction, meaning that under a set of valid axioms, all conclusions drawn through mathematical deduction are certain to be correct. Model superiority stems from selecting the correct prior hypotheses—hypotheses that align with the real world. Additionally, in this process, statistical estimation corresponds to mathematical induction, which is the basis of contemporary machine learning. Based on this process, we proposed hypotheses about the few-shot named entity recognition task, drawing on our background knowledge and understanding. We hypothesize that sample selection bias exists in the task, causing false correlations between the current context and label in each training round. We provide corresponding examples in Figure 1 to illustrate these erroneous correlations. **Building on this hypothesis, we constructed a causal graph and mathematically defined the task (as in Equation 1) and carried out mathematical reasoning (as in Equation 2 and Appendix B). Subsequently, we employed statistical methods to calculate and estimate the probabilities in the equations, resulting in our causal method.**
>
> ​	(2) Following this process, we formally defined the task, provided a mathematical definition based on the causal graph, and derived the equations using causal inference theorems. Therefore, we believe our hypotheses are correct. **In Appendix B, we offer derivations under the causal inference theorem to support our theory.** Simultaneously, our method, based on this hypothesis and causal approach, achieved much higher results than the SOTA, which seems to further substantiate the effectiveness of our method.
>
> ​	(3) Below, we list several published articles on causal inference. Most of these works are based on task understanding, leading to the formulation of their hypotheses and construction of causal models.
>
>   - Niu Y, Tang K, Zhang H, et al. Counterfactual vqa: A cause-effect look at language bias[C]//Proceedings of the IEEE/CVF Conference on Computer Vision and Pattern Recognition. 2021: 12700-12710.
>   - Yang X, Zhang H, Qi G, et al. Causal attention for vision-language tasks[C]//Proceedings of the IEEE/CVF conference on computer vision and pattern recognition. 2021: 9847-9857.
>
> - Wang W, Feng F, He X, et al. Clicks can be cheating: Counterfactual recommendation for mitigating clickbait issue[C]//Proceedings of the 44th International ACM SIGIR Conference on Research and Development in Information Retrieval. 2021: 1288-1297.
>
>
>
> ​	Once again, I appreciate your inquiry. In reality, while causal inference hypotheses and interventions have indeed propelled many tasks forward, the reliability of causal inference does present certain challenges, requiring more research to establish its credibility. We are continuously striving to provide more substantial explanations and evidence. I hope my response addresses your concerns.
>
> 2.The response to 'The section on causal inference is too short and there is no deep elaboration on the connection between this theory and the work proposed in this paper. For example, under which grounds the causal model from Fig 1 is faithful to causal inference? (On the other hand, the authors could clearly eliminate many paragraphs from the paper that are highly repetitive.)
> 	We will further expand the section on causal inference in the subsequent versions and remove redundant content. I'd like to provide a detailed explanation here to clarify the connection between the theory of causal inference and our work.
>
> ​	(1) Regarding the causal graph and Equation 1: In this paper, we start by analyzing the task of few-shot named entity recognition. We have elaborated on this task extensively in the fourth reason for rejection. Through an analysis of the entire process of few-shot named entity recognition, we propose that in each round of sample selection, due to the model's focus on a small subset of the current samples, overfitting to the present data might occur. To address this, we introduce causal intervention to block this path. Building upon this causal graph, we formulate Equation 1. The inspiration for Equation 1 stems from the common concept of front-door adjustment in causal inference. We provide the reference to previous works employing front-door adjustment in the text.
>
>   - Yang X, Zhang H, Qi G, et al. Causal attention for vision-language tasks[C]//Proceedings of the IEEE/CVF conference on computer vision and pattern recognition. 2021: 9847-9857.
>
> ​		(2) **Addressing the faithfulness of the causal model in the graph**: In Appendix B, we present a derivation of Equation 1. The derivation relies on established theorems in causal inference. As long as these theorems are satisfied, the entire derivation process is valid and applicable to both the causal model and causal inference. Allow me to elaborate on the reasoning process. We provide a mathematical definition based on the causal graph, as demonstrated in Equation 1, also shown as Equation 5 in Appendix B. This equation is derived from the law of total probability. By analyzing the structure of the graphical model and assessing variable independence or conditional independence, we deduce that in the path T->E->Y, with T and E as conditions, Y remains unchanged. Therefore, P(Y=y|do(T=t),do(E=e)) and P(Y=y|do(T=t),E=e) are identical. Hence, Equation 5 can be deduced into Equation 6. Moreover, since the causal effect of T->E can be directly obtained from the data without confounding or backdoor paths, it can be derived directly. Consequently, P(E=e|do(T=t)) and P(E=e|T=t) are equivalent, resulting in the derivation of Equation 7. Further, due to the absence of a direct edge between T and Y, intervening on X does not affect Y. Consequently, P(Y = y|do(T = t), do(E = e)) and P(Y = y|do(E = e)) can be deduced as Equation 8. By employing the law of total probability once more and summing over all possible values of T, we obtain Equation 9. Lastly, based on the aforementioned explanation, P(T = t′|do(E = e)) and P(T = t′) are equivalent, resulting in Equation 10. Ultimately, based on the prior explanation, in the path T->E->Y, with T and E as conditions, Y remains unchanged. Therefore, P(Y = y|do(E = e), T = t′) and P(Y = y|E = e, T = t′) are equivalent, leading to Equation 11, also known as Equation 2 in the main text.
>
> ​	(3) **Establishing the connection between our proposed method and Equation 2**: Allow me to offer a comprehensive explanation below. After undergoing causal inference, Equation 2 no longer contains do-operations. Each component of the equation necessitates statistical methods from machine learning to obtain results.
>
> ​	**For the first component, P(E = e|T = t)**, we need to determine the probability of entity e given the context t. Viewing e within its own context renders it a complete, to-be-predicted sample, aligning with the conventional task of prototype-based networks where e corresponds to a specific entity category. This is why we initially employed traditional prototype-based networks to address this part. However, this approach yielded suboptimal results. Upon revisiting this component, we deemed it more appropriate to treat e as a binary classification task rather than associating it directly with specific categories. **We introduced an entity detection method based on this premise.**
>
> ​	 **As for the second component, P(Y = y|E = e, T = t′)**, this signifies the probability of label Y given the current entity e and varying context t'. As t' needs to be iterated through, we maintained the entity e constant while considering the diverse contexts of same-type entities. **This notion gave rise to our context-based causal intervention.** Given the presence of two few-shot settings, namely 5-shot and 1-shot, where 5-shot entails each category having 5 sentences, allowing replacement of same-type contexts, and 1-shot consisting of only one sentence per category with no additional context, we proposed prototype-based causal intervention as an alternative for the latter. Rather than substituting same-type contexts, we considered same-type prototypes. We retained prototypes generated from each round and subsequently combined them when encountering the same-type prototypes. **This approach led to prototype-based causal intervention.**
>
> ​	**Regarding P(T = t′)**, where t' corresponds to context and context influences prediction, we assert that since context impacts prediction, different contexts should not default to equal weights when calculating prototypes. We computed the similarity between the vector representations of support and query data, then assigned distinct weights to samples for prototype computation. **This reasoning underpins the introduction of sample reweighting**.
>
> ​	To conclude, by leveraging our understanding of the task's background knowledge, we formulated hypotheses concerning the causal graph. Subsequently, we mathematically defined the task based on the causal graph and employed causal inference to deduce the equation. Lastly, we proposed different methods to statistically estimate the probability outcomes of various components within the equation.I hope this detailed explanation addresses your queries.
>
> 3.The response to 'Derived from the point above, there is no experimental proof that supports the causal model proposed in this paper. Is the sample selection bias tested? Is there really a causal connection for the form T <- C -> Y? Could the confounder be due to some other biases?
>
> ​	(1) **Regarding the issue of sample selection bias**, our initial skepticism arises from the nature of the few-shot named entity recognition task itself. In a few-shot task, such as a 5-way 5-shot setup, each round of data comprises five classes, with each class containing five sentences. This results in a total of 25 sentences per round (5 × 5 = 25). To learn class-specific features comprehensively, the model tends to heavily fit the patterns within these 25 sentences. The scarcity of samples can lead to overfitting on the current data. If the 25 chosen sentences in a round exhibit significant biases, like the repetition of certain words, the model might inaccurately correlate these occurrences with labels. We are thus concerned that the biased selection of samples could lead to such modeling inaccuracies. To demonstrate this point, we provide an illustration in Figure 1. It can be observed that the presence of the term "square" could spuriously link it to the label "animal." While there might be some connection between "square" and "pigeon," there is no inherent link between "square" and "animal." Additionally, our context-based intervention method, which replaces contexts to alleviate the impact of sample selection bias, results in a notable improvement in accuracy. This suggests that sample selection bias indeed affects the model and our approach effectively addresses it.
>
> ​	(2) **Regarding the depicted relationships in the causal graph**, we have provided a detailed explanation of the causal inference process in your first rejection reason. We simply hypothesize the presence of such a relationship in the real task, and then mathematically define and deduce it based on the causal graph, employing statistical methods for calculation. The entirety of the formula derivation process serves as our theoretical foundation, and the ultimate results validate the effectiveness of our approach.
>
> ​	(3) **Regarding the confounding factors**, in fact, the model encompasses numerous confounding factors, many of which might remain latent or undefined. The sample selection bias that we address in the paper is just one of these factors. We specifically tackle this confounding factor to improve the model and achieve positive outcomes. Future researchers in the field of few-shot named entity recognition are likely to unearth additional factors influencing the model and further enhance its performance. We also aspire to identify more of these confounding factors in future endeavors.
>
> 4.The response to 'The mechanism proposed is derived from Eq. 1, which is unclear if this equation is faithful to the theory of causal inference (no citation nor explanation is given).'
> 	Regarding Eq 1, it is a common form of front-door adjustment often used in causal inference. You can find information about front-door adjustment in "Causal Inference in Statistics: A Primer" by Judea Pearl, Madelyn Glymour, and Nicholas Jewell, Wiley, 2016, or in "Causality: Models, Reasoning and Inference" by Judea Pearl, Cambridge University Press, 2009. Additionally, in the field of computer science, researchers have explored the use of front-door adjustment, as demonstrated in the paper:
>
>  - Yang X, Zhang H, Qi G, et al. Causal attention for vision-language tasks[C]//Proceedings of the IEEE/CVF conference on computer vision and pattern recognition. 2021: 9847-9857.
>
> 5.The response to 'The mechanism proposed consists of some factors, some of which are slightly unclear. For example, for the factor called prototype-based intervention, the intervention done to block the effect of the confounder on the prediction needs the signal from the previous prototype vector. It is unclear what is the previous prototype, what is the exact operation of this intervention, and what is the rationale behind such an intervention. Similarly, for the factor called sample reweighting, it is unclear the logic behind reducing the distribution difference between training and test support data (based on common sense, this operation seems, actually, undesirable since test data should not be used at training time, only at test time).'
> 	I will first explain the entire process of prototype-based few-shot named entity recognition (NER) using the prototype network, then provide a detailed explanation of the prototype-based intervention process and related details, and finally clarify the details of the sample reweighting approach.
>
> ​	(1) In **the process of few-shot NER**, data is divided into training, validation, and test sets. For clarity, let's temporarily disregard the validation set. Both the training and test sets have support data and query data. The support data from the training set is denoted as Strain, and the query data from the training set is denoted as Qtrain. The support data from the test set is denoted as Stest, and the query data from the test set is denoted as Qtest. During training, the model uses Strain to identify the current set of categories and learn their features, then predicts Qtrain data to optimize the model. During testing, the model uses Stest to identify the current set of categories and predicts Qtest data. The obtained loss is not used for model optimization but serves as the final accuracy. In this process, **both the support data from the training set and the support data from the test set are visible to the model to know which categories are present**. The query data from the training set is used for loss prediction and model optimization, while the query data from the test set is only used for prediction. The prototype-based few-shot NER approach refers to training the model's features using Strain (support data from the training set) based on a prototype network. The model averages the vectors of all entities in the same category to create a prototype representation for that category. During query data prediction, the model calculates the distance between the query and all category prototypes, classifying the query into the category with the nearest prototype.
>
> ​	(2) **Our prototype-based intervention involves saving the prototype vectors from each round and using them together in subsequent rounds when encountering the same category.** This means that two prototypes are averaged to obtain a vector representation, rather than considering only the prototype from the current round's data. This intervention stems from the observation of forgetting and overfitting phenomena in the task, as elaborated in the "How does prototype-based causal intervention improve entity recognition" section of the results analysis. If a word (such as "German") is present in the support data of the current round and assigned the category "O," there is a high probability that the model will classify it as "O" in the query data of the current round due to overfitting the support data. However, the correct category for this query data might be "other-language," leading to a model error. Fortunately, in previous rounds, "German" appeared as "other-language." When we introduce the prototype of the "other-language" category from previous rounds to the current round, it helps the model correctly classify "German."
>
> ​	(3) **Regarding sample reweighting**, the traditional prototype network averages all entities in the same category to create a prototype vector as the representation. However, we believe that each entity's contribution to the prototype should not be a default value of 1. If an entity is more similar to the query data, its contribution should be greater. Therefore, we calculate the similarity between the support and query data in the test set, then assign higher weights to the support data with higher similarity when calculating prototypes. It's important to emphasize that we calculate the similarity between support and query data using their vector representations, without considering their category labels, so there is no issue of visibility here. In subsequent prototype calculations, support data with different weights are used to calculate prototypes, and since the support data's weights are calculated based on similarity, it aids in achieving a better match between the distributions of support and query data. This reduces distribution discrepancies, enabling the model to better predict the knowledge learned from support data to query data. This approach aims to enhance model performance by aligning the distributions more effectively.
>
>
>
> **Q1.** In the tables of results, do +- indicate standard deviations across different random seeds?
> 	The observed discrepancies in the results are attributed to the influence of random seed initialization. Indeed, these fluctuations stem from the random seed settings, thereby impacting the outcome. Moreover, to ensure the reproducibility and non-random nature of the outcomes, we conducted multiple experimental runs and subsequently calculated the average values.
>
> **Q2.** There seem to be 2 mistakes in Table 2: on the Mu and Se domains, the model with highest score is L-TapNet+CDT and not the model proposed in this paper, is this correct?
> 	Table 2 is indeed accurate. The experimental findings highlight that our outcomes might not be optimal for these two specific categories. However, considering the results across the other categories and amalgamating them, we presented the "Avg" figure. **On the aggregate scale, while these two categories may exhibit slight deficits, the overall average surpasses the state-of-the-art (SOTA) benchmarks.** This substantiates the efficacy of our proposed approach.
>
>  - Yang X, Zhang H, Qi G, et al. Causal attention for vision-language tasks[C]//Proceedings of the IEEE/CVF conference on computer vision and pattern recognition. 2021: 9847-9857.
>  - Niu Y, Tang K, Zhang H, et al. Counterfactual vqa: A cause-effect look at language bias[C]//Proceedings of the IEEE/CVF Conference on Computer Vision and Pattern Recognition. 2021: 12700-12710.
>  - Chang C H, Adam G A, Goldenberg A. Towards robust classification model by counterfactual and invariant data generation[C]//Proceedings of the IEEE/CVF Conference on Computer Vision and Pattern Recognition. 2021: 15212-15221.

---

### Official Review · Reviewer_djTo · 2023-08-04

**Typos Grammar Style And Presentation Improvements:** 1.	The sentence “The Softmax and MMD …
**Soundness:** 3

**Excitement:**

3: Ambivalent: It has merits (e.g., it reports state-of-the-art results, the idea is nice), but there are key weaknesses (e.g., it describes incremental work), and it can significantly benefit from another round of revision. However, I won't object to accepting it if my co-reviewers champion it.

**Paper Topic And Main Contributions:**

This paper aims to solve the overfitting problem (i.e., superficial correlation resulting from the data bias) of the few-shot NER task. It proposes a causal intervention framework to block the backdoor between the context and the label. Specifically, the authors introduce the components including entity detection, context-based intervention and sample reweighting. For the one-shot scenario, they replace the context-based intervention by a prototype-based one considering the lack of additional samples. The experiments on two datasets also demonstrate the effectiveness of the proposed method.

**Questions For The Authors:**

1.	How does the sample reweighting component reflect the expression P(T=t’) in Eq. 2?
2.	How to understand the impact of prototype-based causal intervention on data bias? Could you provide some examples, since I think this is not as intuitive as context-based causal intervention.
3.	Why not use the same baselines on both two datasets?
4.	Could you provide the intra-class distance and inter-class distance respectively for the t-SNE visualization in Fig. 4, since I think qualitative analysis is more persuasive.
5.	What do the horizontal and vertical axes in Figure 5 respectively represent?
6.	Why does causal intervention show much better results when entity detection is performed? What would be the outcome if the entity detection component is added to other baselines?

**Reasons To Accept:**

1. This paper focuses on the problem of data bias and model generalization in the few-shot NER task, which is a meaningful topic.
2. The experimental results show the significant improvement over state-of-the-art baselines.

**Reasons To Reject:**

1. The novelty is limited. A causal-intervention based framework is commonly used to overcome superficial correlation in datasets, which I think lacks innovation in terms of methodology.

2. In the experiments, the latest LLM, e.g., ChatGPT is not compared. In addition, it can be observed that the role of entity detection in the performance improvement is significant, but its relevance to the perspective of causal intervention in this paper is not particularly strong. Similarly, it is difficult for me to ascertain whether certain implementation details have a significant impact on the model performance, such as incorporating MMD loss into the loss function or performing fine-tuning using the support instances from the test data.

3. The paper does not elaborate on the overall architecture and key details of the model. For example, how to bridge the different components and how to perform entity detection are not clearly illustrated. I suggest providing more precise formalized descriptions.

**Reproducibility:**

4: Could mostly reproduce the results, but there may be some variation because of sample variance or minor variations in their interpretation of the protocol or method.

**Reviewer Confidence:**

3: Pretty sure, but there's a chance I missed something. Although I have a good feel for this area in general, I did not carefully check the paper's details, e.g., the math, experimental design, or novelty.

---

> ### Author Rebuttal · Authors · 2023-08-29
>
> Thank you for taking the time to review our paper. We appreciate your insightful comments and valuable feedback, which have undoubtedly improved the quality of our paper. For your review, we would like to address each of your points in detail.
>
> 1.The response to 'The novelty is limited. A causal-intervention based framework is commonly used to overcome superficial correlation in datasets, which I think lacks innovation in terms of methodology'.
>
> Regarding this question, we provide the following explanation.
>
> ​	(1)To begin with, causal intervention is indeed widely employed to address spurious correlations within tasks. **However, within the causal framework, the methods proposed in each paper differ due to the varying nature of tasks and causal graphs.** While our approach is rooted in the framework of causal intervention, the diversity in both the underlying causal intervention concepts and the tasks themselves leads to unique methodologies in each study. We also provide references to a few papers grounded in causal intervention below. Despite sharing the foundation of causal intervention, each study's methodology remains distinctive and innovative.
>
> - Yang X, Zhang H, Qi G, et al. Causal attention for vision-language tasks[C]//Proceedings of the IEEE/CVF conference on computer vision and pattern recognition. 2021: 9847-9857.
>
> - Zhang W, Lin H, Han X, et al. De-biasing distantly supervised named entity recognition via causal intervention[J]. arXiv preprint arXiv:2106.09233, 2021..
>
> - Wu X, Li C, Miao Y. Discovering topics in long-tailed corpora with causal intervention[C]//Findings of the Association for Computational Linguistics: ACL-IJCNLP 2021. 2021: 175-185.
>
> ​	(2) In terms of the innovation of our paper, there are several key aspects to consider. Firstly, in the entity detection part, we depart significantly from traditional prototype networks and existing methods. **We are the first to consider that all entity categories in few-shot named entity recognition should exhibit commonalities that differentiate them from non-entities.** Therefore, we introduce a novel approach of categorizing all entity classes into a single overarching class for input into the prototype network. Furthermore, our introduction of sample reweighting represents a departure from previous works in few-shot named entity recognition, which typically assumed equal weights of 1 for all samples during prototype computation. **We are the first to introduce the consideration of sample weights in this task.**
>
> 2.The response to 'In the experiments, the latest LLM, e.g., ChatGPT is not compared. In addition, it can be observed that the role of entity detection in the performance improvement is significant, but its relevance to the perspective of causal intervention in this paper is not particularly strong. Similarly, it is difficult for me to ascertain whether certain implementation details have a significant impact on the model performance, such as incorporating MMD loss into the loss function or performing fine-tuning using the support instances from the test data'.
>
> ​	(1) Below, we list articles and results related to using GPT for few-shot NER tasks:
>
> ​			Ashok D, Lipton Z C. PromptNER: Prompting For Named Entity Recognition[J]. arXiv preprint arXiv:2305.15444, 2023.
>
> （Supplement Table D1: Results in FewNERD-intra dataset 10-way 5-shot)
>
> | F1（%） | FewNERD-intra |
> | ------- | ------------- |
> | GPT3    | 62.33         |
> | GPT4    | 72.63         |
> | Ours    | 61.33         |
>
> The table indicates that our results are comparable to GPT-3 but fall short of the performance achieved by GPT-4. However, it's important to note that our experimental execution time is significantly shorter when compared to utilizing GPT-4 in each data round. Additionally, our model has a smaller parameter size compared to GPT4, contributing to its efficiency.At the same time, GPT-4 also comes with a high economic cost.
>
> ​	(2) The following table presents the ablation results for each component, illustrating that each component has a substantial impact on the task performance.
>
> (Supplement Table D2): Results in FewNERD-intra dataset. w. and w.o. mean using the component or not.
>
> | F1（%）                 | FewNERD-intra |
> | ----------------------- | ------------- |
> | w.o. context-based      | 60.44         |
> | w context-based         | 70.34         |
> | w.o. prototype-based    | 46.09         |
> | w prototype-based       | 54            |
> | w.o. sample reweighting | 50            |
> | w sample reweighting    | 54            |
> | w.o. entity detection   | 62.8          |
> | w entity detection      | 70.34         |
>
> ​	Entity detection can assist the model in performing initial binary classification on sequences, facilitating more accurate classification in subsequent context-based causal intervention.
>
> ​	(3)Regarding your elaboration on the effects of MMD loss and fine-tuning on the results: **In the 5-shot scenario, we did not include the MMD loss, yet we still achieved promising results.**
>
> (Supplement Table D3): Results in FewNERD-intra and FewNERD-inter dataset 5-way 5-shot.
>
> | F1（%） | FewNERD-intra | FewNERD-inter |
> | ------- | ------------- | ------------- |
> | SOTA    | 63.23         | 74.14         |
> | Ours    | 70.34         | 81.89         |
>
> Additionally, we extended our experiments in this direction by omitting the MMD loss and adopting the conventional loss.
>
> (Supplement Table D4): Results in FewNERD-intra and FewNERD-inter dataset 5-way 1-shot. w. and w.o. mean using MMD or not.
>
> | F1（%）  | FewNERD-intra | FewNERD-inter |
> | -------- | ------------- | ------------- |
> | SOTA     | 52.04         | 68.77         |
> | w.o. MMD | 52.97         | 69.15         |
> | w. MMD   | 54            | 69.41         |
>
> ​	**It can be observed that the impact of MMD on the results is not significant**.
>
> ​	(4) Concerning fine-tuning, **it's a common practice in the realm of few-shot learning and is often employed in various few-shot named entity recognition tasks**. For example, I have listed below two references that have been accepted by ACL. **It's undeniable that fine-tuning can indeed yield improvements in results. However, our approach combined with fine-tuning led to even greater enhancements.**
>
> - Das S S S, Katiyar A, Passonneau R J, et al. CONTaiNER: Few-shot named entity recognition via contrastive learning[J]. arXiv preprint arXiv:2109.07589, 2021.
> - Ma T, Jiang H, Wu Q, et al. Decomposed meta-learning for few-shot named entity recognition[J]. arXiv preprint arXiv:2204.05751, 2022.
>
> 3.The response to 'The paper does not elaborate on the overall architecture and key details of the model. For example, how to bridge the different components and how to perform entity detection are not clearly illustrated. I suggest providing more precise formalized descriptions'.
>
> ​	Due to webpage constraints, it seems that we are currently unable to display the complete model diagram we have prepared. We will include this section in subsequent versions. For now, let me provide a brief description of the overall process. Within each round of data, there are support and query sets.
>
> ​	**In the entity detection component**, for the support data, we aggregate all entity classes into a single major category, denoted as the "class prototype" in the diagram. Simultaneously, all non-entity classes are grouped into another major category, corresponding to the "other prototype" in the diagram. Subsequently, adhering to the conventional prototype network methodology, we compute prototypes for both of these categories and employ the query data to calculate distances. In this phase, each query yields a two-dimensional outcome, signifying the probabilities for it belonging to the entity and non-entity classes.
>
> ​	Following this, **in the causal intervention segment**, for instances with 5-shot, we engage context-based intervention. For 1-shot instances where contextual replacement is unfeasible, we employ prototype-based intervention. Detailed depictions of context-based and prototype-based interventions are provided in the paper with illustrative diagrams. Through intervention, the dimensionality of the outcomes for each query becomes dependent on the number of categories, denoted as N. Resulting from intervention, the probability distribution is obtained for each category.
>
> ​	**In the section involving sample reweighting**, the similarity between each support data and query data is computed. This similarity calculation informs the weight assigned to the support data during prototype computation. Traditional prototype network computation is then conducted using these assigned weights. In this phase, akin to the previous stage, each query generates results with dimensions corresponding to the number of categories, N.
>
> ​	Finally, **during the concatenation of results**, due to the distinct dimensions of outcomes from entity detection compared to the other two components, we replicate the probabilities of entity classes the same number of times as N, mirroring the diagram's depiction. Upon achieving parity in dimensions, the results from the three components are summed element-wise to yield the overall outcome.
>
> **Q1.** The response to 'How does the sample reweighting component reflect the expression P(T=t’) in Eq. 2'.
>
> ​	In this equation, T represents the context, and T=t' signifies the probability corresponding to different contextual scenarios. We posit that distinct contexts exert varying influences on the outcome. Therefore, we endeavor to assign differing levels of importance to different contexts. To achieve this, we leverage the similarity between the current sample and the query as a weighting factor, aimed at distinguishing the significance of various contexts in prototype computation.
>
> **Q2.** The response to 'How to understand the impact of prototype-based causal intervention on data bias? Could you provide some examples, since I think this is not as intuitive as context-based causal intervention'.
>
> ​	In the results analysis section of the paper, titled "How does prototype-based causal intervention improve entity recognition," you provide examples to illustrate the impact of prototype-based intervention. Due to the webpage's configuration, it appears that we are unable to display the properly formatted example diagram at the moment. We will incorporate this section in the upcoming versions. Here, let me briefly describe the illustration. Regarding the term "German," if it is included in the current support data and categorized as the "O" class, then in the context of the current query data, a conventional prototype network would classify it as the "O" class as well. This illustrates that the model is only fitting to the immediate data and is forgetting previous data. However, when we incorporate prior knowledge, if the term "German" appeared in support data in previous rounds and was labeled as the "other-language" class, the model can correctly classify it as the "other-language" category. This is due to the fact that the previous prototypes retain relevant feature knowledge, which we reintroduce into the current scenario.
>
> **Q3.** The response to 'Why not use the same baselines on both two datasets‘.
> 	The issue with the baseline models in the context of the datasets arises from the disparate timeframes of their introduction. FewNERD dataset was introduced at a later point. Consequently, the baseline model count is lower for FewNERD compared to the other dataset. We conducted comparisons based on the initial state-of-the-art (SOTA) baselines available on Paperwithcode. Additionally, certain baseline models were exclusively tested on one of the datasets, and we directly referenced these outcomes.
>
> **Q4.** The response to 'Could you provide the intra-class distance and inter-class distance respectively for the t-SNE visualization in Fig. 4, since I think qualitative analysis is more persuasive‘.
>
> ​	 I'm not quite clear on whether you're referring to the intra-class and inter-class distances in terms of vector representations from the FewNER-intra dataset and the FewNERD-inter dataset, or if you mean the distances within each class and the distances between different classes in the vector representations obtained from any given dataset.
>
> ​	(1) If you are referring to the first scenario, involving vector representations from the FewNER-intra dataset and the FewNERD-inter dataset: We have provided the vector representations from the FewNERD-intra dataset in the paper. Here, we are presenting the vector representations from the FewNERD-inter dataset. (Due to webpage constraints, it seems that we are unable to display the completed diagrams. We will include this section in future versions. Here, I will provide a brief description of the diagrams.) The five color categories in the diagram represent five different classes. It can be observed that in a regular network, the classes are scattered and intertwined, making separation difficult. However, upon applying our method, the boundaries between classes become distinct, and instances of the same class start to cluster together.
>
> ​	(2) If your query pertains to the second scenario, involving distances within each class and distances between different classes in the vector representations obtained from any given dataset: We have already presented the vector representations from the FewNERD-intra dataset in the paper. The five colors represent five distinct classes. It can be noted that distances within the same color group have become closer, and there are evident boundaries between different colors.
>
> **Q5.** The response to 'What do the horizontal and vertical axes in Figure 5 respectively represent'.
>
> ​	Regarding this matter, I'd like to provide some further elucidation. Figure 5 manifests as a histogram, serving the purpose of delineating the congruity within data distribution across the task.  **Conventionally, the horizontal axis signifies distinct data types, while the vertical axis encapsulates the corresponding distribution magnitude. **
>
> **Q6**. The response to 'Why does causal intervention show much better results when entity detection is performed? What would be the outcome if the entity detection component is added to other baselines'.
>
> ​	 (1) To begin with, our introduced entity detection indeed has significantly contributed to enhancing the task. It enables the model to identify non-entity tokens. Under this premise, causal intervention can manifest its effectiveness more profoundly. **In traditional prototype networks, they encompass two responsibilities: distinguishing entities from non-entities and further categorizing entities into specific classes. However, our entity detection solely requires the segregation of entities from non-entities, deferring the specific classification to subsequent stages of the task.** Thus, entity detection aids the model in preliminary screening and filtering, ultimately leading to improved performance.
>
> ​	(2) The incorporation of entity detection into other baselines isn't unprecedented. **Prior works have explored the concept of detecting entities and subsequently categorizing them, often termed as "span" methods. We provide references to two such studies below. Evidently, incorporating span-based approaches can elevate model performance, yet our results surpass the state-of-the-art (SOTA).** This substantiates the efficacy of our proposed causal intervention method.
>
> - Wang P, Xu R, Liu T, et al. An enhanced span-based decomposition method for few-shot sequence labeling[J]. arXiv preprint arXiv:2109.13023, 2021.
>
> - Ma T, Jiang H, Wu Q, et al. Decomposed meta-learning for few-shot named entity recognition[J]. arXiv preprint arXiv:2204.05751, 2022
>
>
>
> ​	Finally, we have provided detailed explanations for your suggestions and questions. We sincerely appreciate your valuable feedback and inquiries, as they have helped us enhance our paper. We hope that our responses have addressed your concerns. If there are any remaining points that we haven't clarified, please feel free to ask again.

---

### Meta-Review · Area_Chair_W1jm · 2023-09-19

**Recommendation:** 4

**Metareview:**

All reviewers agreed that the proposed approach for few-shot NER is sound and useful. The results on two benchmarks show clear benefits, some surpassing current results in the existing literature.

However, there are some raised legitimate concerns.
In particular, reviewers are concerned about the lack of clarity in the explanation of the causal model and details on the model component (e.g., sample reweighting component), some of which are partially addressed during the rebuttal phase. It is suggested the authors provide explicit connections to existing causal intervention approaches and use more case studies and visualizations to illustrate their approach.

Regarding presentation, the reviewers pointed out several typos, grammatical errors, and inconsistencies in the paper that need to be addressed.

---

### Decision · Program_Chairs · 2023-10-07

**Decision:**

Accept-Findings

**Comment:**

All reviewers agreed that the proposed approach for few-shot NER is sound and useful. The results on two benchmarks show clear benefits, some surpassing current results in the existing literature.

However, there are some raised legitimate concerns.
In particular, reviewers are concerned about the lack of clarity in the explanation of the causal model and details on the model component (e.g., sample reweighting component), some of which are partially addressed during the rebuttal phase. It is suggested the authors provide explicit connections to existing causal intervention approaches and use more case studies and visualizations to illustrate their approach.

Regarding presentation, the reviewers pointed out several typos, grammatical errors, and inconsistencies in the paper that need to be addressed.